# Why Do Ferroelectrics Exhibit Negative Capacitance?

**DOI:** 10.3390/ma12223743

**Published:** 2019-11-13

**Authors:** Michael Hoffmann, Prasanna Venkatesan Ravindran, Asif Islam Khan

**Affiliations:** 1NaMLab gGmbH/TU Dresden, 01187 Dresden, Germany; michael.hoffmann@namlab.com; 2School of Electrical and Computer Engineering, Georgia Institute of Technology, Atlanta, GA 30332, USA; rprasanna22@gatech.edu; 3School of Materials Science and Engineering, Georgia Institute of Technology, Atlanta, GA 30332, USA

**Keywords:** ferroelectricity, negative capacitance, polarization catastrophe

## Abstract

The Landau theory of phase transitions predicts the presence of a negative capacitance in ferroelectric materials based on a mean-field approach. While recent experimental results confirm this prediction, the microscopic origin of negative capacitance in ferroelectrics is often debated. This study provides a simple, physical explanation of the negative capacitance phenomenon—i.e., ‘S’-shaped polarization vs. electric field curve—without having to invoke the Landau phenomenology. The discussion is inspired by pedagogical models of ferroelectricity as often presented in classic text-books such as the *Feynman lectures on Physics* and *the Introduction of Solid State Physics* by Charles Kittel, which are routinely used to describe the quintessential ferroelectric phenomena such as the Curie-Weiss law and the emergence of spontaneous polarization below the Curie temperature. The model presented herein is overly simplified and ignores many of the complex interactions in real ferroelectrics; however, this model reveals an important insight: The polarization catastrophe phenomenon that is required to describe the onset of ferroelectricity naturally leads to the thermodynamic instability that is negative capacitance. Considering the interaction of electric dipoles and saturation of the dipole moments at large local electric fields we derive the full ‘S’-curve relating the ferroelectric polarization and the electric field, in qualitative agreement with Landau theory.

## 1. Introduction

Ferroelectric materials possess a spontaneous polarization that can be reversed by the application of an electric field. Because of their unique properties, ferroelectrics have been considered for a wide variety of applications ranging from infrared detection to non-volatile information storage. Recently, the use of negative capacitance in ferroelectrics was proposed to overcome the fundamental limits of power dissipation in integrated circuits [1]. However, this has led to a lot of debate on the origin and feasibility of negative capacitance in ferroelectric materials. In particular, so far, most investigations on negative capacitance have utilized the Landau theory of ferroelectric phase transitions without giving an explanation from the microscopic point of view. This has often raised the question whether negative capacitance is an unphysical, artificial construct for the convenience of the phenomenology of Landau theory. The goal of this work is to show that negative capacitance naturally arises in ferroelectrics based on an easily understandable, microscopic model.

The spontaneous polarization in a ferroelectric originates from its non-centrosymmetric crystal structure, below the Curie-temperature $T_{C}$, which leads to a permanent electric dipole moment, even when no electric field is applied. For temperatures above $T_{C}$, the material transitions into a paraelectric phase, which has no spontaneous polarization. These ferroelectric phase transitions have been well understood for more than 70 years based on Landau’s phenomenological theory of phase transitions [2]. Landau’s theory takes a thermodynamic mean-field approach to analyze a phase transition based on symmetry considerations only. Ferroelectrics were first described using the Landau framework in the works of Ginzburg [3] and Devonshire [4] in the 1940s.

In these seminal works, the relationship between the ferroelectric polarization *P* and the electric field *E* is given by an ‘S’-shaped curve, which is the result of a symmetric double-well free energy landscape. Since this ‘S’-shaped *P*-*E* curve has a region of negative slope d*P*/d*E* < 0, it was originally argued by Landauer, that ferroelectric should possess a negative capacitance, which could in principle be stabilized (see Figure 1) [5]. However, in most other works at the time, this region of negative slope was only described as "thermodynamically unstable" and therefore not accessible in experiments (see Figure 2). Only recently, first experimental evidence of negative capacitance in ferroelectrics has been reported [6,7,8,9,10]. However, since Landau’s framework presents a mean-field theory, which explains the observed macroscopic phenomenon only, an intuitive microscopic understanding of the origin of negative capacitance in ferroelectric materials is needed.

Feynman presented a pedagogical approach to explain the microscopic origin of ferroelectricity in his classic *lectures on physics* [14]. However, he did not discuss the resulting negative sign in the polarization-electric field dependence, besides mentioning the presence of a polarization catastrophe which leads to a runaway condition with the dipole moments increasing to infinity, which he described as implausible. Kittel, on the other hand, presented a more mathematically rigorous model based on the Clausius-Mossotti relation based on a similar initial framework [15].

In contrast, most discussions in the field of negative capacitance have so far only utilized Landau’s phenomenological mean-field framework and did not investigate the actual, microscopic origin of the effect. Recently, in ref. [16], Wong and Salahuddin presented a model for the origin of the ‘S’-shaped *P*-*E* curve of a ferroelectric unit cell based on dipole-dipole and dipole-electric field interactions. This work also discusses the stabilization of a negative capacitor through a depolarization field caused by reduced charge screening by adding a positive capacitance in series. Furthermore, ref. [17] presented a new perspective on negative capacitance relating it to the microscopic mechanism for the onset of the polarization catastrophe.

In the following, the simple pedagogical model proposed by Feynman is used to analyze the behaviour of a ferroelectric material below the Curie-temperature. A mathematical model for the relation between the electric dipole moment *p* and local electric field at the dipole $E_{local}$ is presented, which reveals the microscopic origin of negative capacitance in ferroelectrics. This model is then extended to qualitatively reproduce the ‘S’-shaped *P*-*E* curve and the double-well energy landscape known from the phenomenological Landau mean-field theory [18].

## 2. Model Description

### 2.1. Feynman’s Pedagogical Model

Consider a one-dimensional model where each ferroelectric unit cell is modeled as an electric dipole. Let the model be an infinite line of dipoles with dipole moment *p* equally spaced with a lattice constant ‘*a*’. Since this model assumes an infinitely long lattice, each dipole has the same dipole moment *p* and experiences the same electric field.

The dipole moment is dependent on the local electric field at the dipole. $E_{local}$ is the sum of the applied electric field and the electric field due to all other dipoles in the system. When an electric field $E_{applied}$ is present, the dipole moment of each dipole is given by the following relations.

(1)$$p=\alpha \epsilon_{0}E_{local}$$

(2)$$E_{local}=E_{dipole}+E_{applied}$$

Here, α is the linear polarizability of the material and $\epsilon_{0}$ is the permittivity of free space. From electrostatics, the field due to a dipole at a distance *r* from its origin is given by $E_{r}=\frac{1}{4\pi \epsilon_{0}}\frac{2p}{a^{3}}$. Based on the one-dimensional model, $E_{dipole}$ at each dipole is simply the sum of the fields due to all dipoles and can be written as an infinite series.

(3)$$E_{dipole}=2\frac{1}{4\pi \epsilon_{0}}\frac{2p}{a^{3}}\left(1+\frac{1}{2^{3}}+\frac{1}{3^{3}}+\ldots \right)=\frac{p}{\epsilon_{0}}\frac{0.383}{a^{3}}$$

This shows that $E_{dipole}$ is directly proportional to the dipole moment of the dipoles. To simplify Equation (Equation 3), a structural factor ζ is introduced which depends on the geometry of the system. ζ takes different values depending on the microscopic arrangement of the dipoles in the lattice. In the discussion to follow, ζ is treated as a variable used for dimensional correctness. Therefore, we can write

(4)$$E_{dipole}=\zeta p$$

From Equations (1), (2) and (4), we then obtain

(5)$$p=\frac{\alpha \epsilon_{0}}{1-\alpha \epsilon_{0}\zeta }E_{applied}$$

In the Feynman model, the dipole moment *p* is proportional to the local electric field $E_{local}$, which in turn is dependent on *p* thereby resulting in a microscopic feedback loop. For positive values of α and ζ, Equation (Equation 5) describes positive feedback.

The dielectric susceptibility, $\chi_{e}$ is proportional to the rate of change of the dipole moment *p* with respect the applied electric field $E_{applied}$.

(6)$$\chi_{e}=\frac{1}{\epsilon_{0}}\frac{dp}{dE_{applied}}=\frac{\alpha }{1-\alpha \epsilon_{0}\zeta }$$

The polarizability α is generally inversely proportional to the absolute temperature *T*. Let $T_{C}$ be the Curie temperature of the material. For $T>T_{C}$, it follows that $\alpha \epsilon_{0}\zeta <1$ and $\chi_{e}$ is positive. The material exhibits paraelectric behaviour. According to the Curie-Weiss Law, the electric susceptibility $\chi_{e}$ is inversely proportional to $\left(T-T_{C}\right)$. The constant of proportionality *C* is the Curie constant.

(7)$$\chi_{e}=\frac{C}{T-T_{C}}$$

At the Curie temperature, $\frac{1}{\alpha \epsilon_{0}}=\zeta$, leading to a singularity of $\chi_{e}$. The interest of this study is in the state of the system below $T_{C}$. For $T<T_{C}$, $\frac{1}{\alpha \epsilon_{0}}<\zeta$ which gives rise to a negative susceptibility. Since the relative permittiviy for a ferroelectric can be approximated by $\epsilon_{r}=1+\chi_{e}\approx \chi_{e}$, this indicates that ferroelectric materials intrinsically exhibit negative permittivity and thus negative capacitance.

A common misconception is that, in the negative capacitance region, the ferroelectric dipoles are aligned in the direction opposite to that of the electric field, which would be a violation of the fundamental laws of thermodynamics because the anti-parallel alignment corresponds to an energy maximum. However, as Equation (Equation 1) shows, the dipole moments always points in the same direction as the *local* electric field.

### 2.2. Relation between *p* and $E_{applied}$

The Feynman model of ferroelectrics which we discussed so far would thus allow the system to spontaneously polarize for even the smallest deviation of either *p* or $E_{applied}$ from zero. This is due to the relation between *p* and *E* in Equation (Equation 1), describing a positive feedback loop. Indeed, from the discussion so far, the polarization would increase to infinity without bound. Historically, this effect has been called the polarization catastrophe, which is at the origin of the intrinsic negative susceptibility derived here.

To explain why the polarization eventually saturates and does not diverge to infinity, we have to introduce the polarizability as a non-linear function of the local electric field. In general, the application of an electric field increases the distance between the positive and negative ions of a dipole, thus increasing its dipole moment. However, the distance only increases until the attractive forces in the bond between the opposite charged ions balance the force due to the local electric field. Let $p_{max}$ correspond to the saturation dipole moment and let $E_{critical}$ be the local electric field at which the dipole moment starts to saturate. For simplicity, let us assume the following relation between *p* and $E_{local}$.

(8)$$p=p_{max}\tanh\left(\frac{E_{local}}{E_{critical}}\right)$$

For small values of $E_{local}$, that is, when $E_{local}<<E_{critical}$, *p* and $E_{local}$ are related linearly as in Equation (Equation 1). Combining Equations (2), (4) and (8), the following expression is obtained.

(9)$$p=p_{max}\tanh\left(\frac{\zeta p+E_{applied}}{E_{critical}}\right)$$

Using the identity, $\tanh^{-1}=\frac{1}{2}\log\frac{1+x}{1-x}$ in Equation (Equation 9), we can then write

(10)$$E_{applied}=-\zeta p+\frac{E_{critical}}{2}\log\frac{1+p/p_{max}}{1-p/p_{max}}$$

As we have established before, the linear polarizability α is inversely proportional to the temperature *T*. Let $\alpha =\frac{1}{\sigma \epsilon_{0}T}$, where σ is a positive constant and $\zeta =\sigma T_{C}$. Equation (Equation 10) simplifies to the following.

(11)$$\frac{E_{applied}}{E_{critical}}=-\frac{T_{C}}{T}\frac{p}{p_{max}}+\frac{1}{2}\log\frac{1+p/p_{max}}{1-p/p_{max}}$$

Several *p*-$E_{applied}$ curves for different temperatures $T/T_{C}$ are shown in Figure 3. The “S”-shaped *p*-*E* curve is seen for $T<T_{C}$ indicating the presence of negative susceptibility. This is in qualitative agreement with the polarization-electric field curves obtained from homogeneous Landau theory.

Using the Taylor expansion in Equation (Equation 10) in the limits, $|E_{applied}|<E_{critical}$ and $|p|<p_{max}$, we obtain

(12)$$\begin{array}{c} \begin{array}{cc} E_{applied} & =-\zeta p+E_{critical}\left(\frac{p}{p_{max}}+\frac{1}{3}{\left(\frac{p}{p_{max}}\right)}^{3}+\frac{1}{5}{\left(\frac{p}{p_{max}}\right)}^{5}+\ldots \right)\\ \; & =\left(\frac{E_{critical}}{p_{max}}-\zeta \right)p+\left(\frac{E_{critical}}{3p_{max}^{3}}\right)p^{3}+\left(\frac{E_{critical}}{5p_{max}^{5}}\right)p^{5}+\ldots \end{array} \end{array}$$

We can now relate this expression for the microscopic dipole moments *p* to the macroscopic polarization *P*, which is defined as the average dipole moment per unit volume *V*, i.e. P=p/V. The well-known mean-field Landau formalism is given by $E_{applied}=a_{1}P+a_{11}P^{3}+a_{111}P^{5}$, which means that we can now identify the Landau-coefficients $a_{1}$, $a_{11}$ and $a_{111}$ by comparison to Equation (Equation 12). This yields $a_{1}=V\left(E_{critical}/p_{max}-\zeta \right)$, $a_{11}=V\left(E_{critical}/3p_{max}^{3}\right)$ and $a_{111}=V\left(E_{critical}/5p_{max}^{5}\right)$.

In a next step, we can calculate the electrostatic potential energy *U* of one electric dipole, which can be generally defined as U=∫Edp. When we integrate the applied field with respect to the dipole moment *p* from Equation (Equation 11), the energy of the system can be written as follows. 

(13)$$\begin{array}{c} \begin{array}{cc} U & =\frac{E_{critical}}{2}\left[-\frac{T_{C}}{T}\frac{p^{2}}{p_{max}}+p_{max}\log\left(1-p^{2}/p_{max}^{2}\right)+p\log\left(\frac{p_{max}+p}{p_{max}-p}\right)\right] \end{array} \end{array}$$

Here the integration constant of the energy integral was set to zero. Figure 4 shows the energy profiles at different temperatures. For $T<T_{C}$, a double-well energy landscape emerges, indicating the presence of two degenerate stable states corresponding to a spontaneous dipole moment.

As can be seen from Figure 4 for $T/T_{C}<1$, the unpolarized state at $p/p_{max}=0$ is unstable, leading to an increase in dipole moment even through thermal fluctuations and is bounded by the saturation of the dipole moment for high electric fields. Negative capacitance in ferroelectrics thus originates from the polarization catastrophe below $T_{C}$, which leads to the emergence of the spontaneous polarization itself.

## 3. Conclusions

An intuitive, microscopic description of the phenomenon of negative capacitance in ferroelectrics has been presented. It was shown that a positive feedback mechanism operates to align the electric dipole moments in the direction of the local electric field, which may also be in the direction opposite to the applied electric field, leading to a negative capacitance without any violation of fundamental thermodynamics. The positive feedback is set up because the electric field caused by the dipoles is larger than the applied electric field. The spontaneous electric dipole moment is bounded by a negative feedback loop due to attractive bonding forces in the atoms and the interaction with the electric field. By assuming a non-linear relation between the dipole moment *p* and the local electric field $E_{local}$ such that *p* saturates at large values of $E_{local}$, the ‘S’-curve relating the polarization *P* and the applied electric field $E_{applied}$ was derived. The same analysis leads to the double-well ferroelectric energy profile.

It needs to be noted that the model presented herein is a toy model in one dimension, and does not represent any real ferroelectric material in three dimensions. As Feynman pointed out, having multiple instances of such one-dimensional chains close by in parallel will actually lead to antiferroelectricity, not ferroelectricity. An appropriate treatment of ferroelectrics negative capacitance in three dimensions was presented in ref. [16]. However, even such a simplified, one dimensional model provides a powerful insight: The polarization catastrophe phenomenon that is required to describe displacement type ferroelectrics naturally leads to a thermodynamically unstable, negative capacitance.

Furthermore, it is rather intriguing that the single domain picture as originally proposed [1] and also utilized in this work plays a significant role in explaining negative capacitance in ferroelectrics while experiments demonstrate the existence of this phenomena in multi-domain materials. Ferroelectrics—even in the widely studied, archetypal ones in their cleanest and highest quality, epitaxial forms—are indeed complicated materials; in fact, complexities therein span multiple orders of length scale. For example, in ferroelectric-dielectric heterostructures, the very same depolarizing field that stabilizes the otherwise unstable negative capacitance in the ferroelectric causes it to decompose into complicated domain structures [8,9]. Even in such a multi-domain scenario, negative capacitance states are experimentally observed in nanoscale regions within the ferroelectric layer [8]. On the other hand, recent pulsed capacitance measurements of ferroelectric-dielectric heterostructures have led to an experimental validation of the hysteresis-free ‘S’-shaped polarization-electric field relation in the ferroelectric [7,19,20]. This points to the fact that our current understanding is not adequate in explaining the full spectrum of negative capacitance phenomena. Furthermore, depolarization phenomena are of interest since they can limit the use of ferroelectrics for memory applications by affecting the retention of the device, while the same physical effect opens up opportunities for the design of negative capacitance devices [21]. While these topics are beyond the scope of this work, an overview of the current status of negative capacitance transistors was recently presented in ref. [22,23,24]. Furthermore, a detailed analysis of ferroelectric negative capacitance phenomena in the multi-domain scenario has recently been presented in ref. [25], which interested readers are encouraged to peruse through.

Besides the recent progress in the basic theoretical understanding of negative capacitance, further investigations are necessary to prove that negative capacitance effects can also be utilized in application-relevant ferroelectric materials based on HfO${}_{2}$ and ZrO${}_{2}$ [26,27]. In this relatively new class of ferroelectrics, non-ideal effects like their polycrystalline morphology, crystal defects and charge trapping phenomena might play an important role in the design and operation of practical negative capacitance devices [28]. Therefore, future experimental and theoretical work should also focus on moving from idealized ferroelectric model systems (e.g., epitaxial perovskite superlattices) towards more application-relevant ferroelectric materials and devices.

## Figures and Tables

**Figure 1 materials-12-03743-f001:**
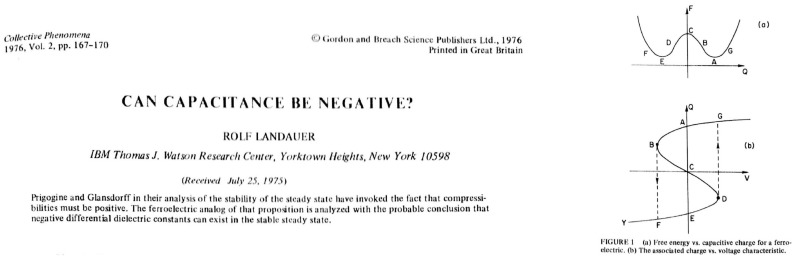
After the discovery of ferroelectricity in $BaTiO_{3}$ in the 1940s, Landauer presented the ‘S’-curve relation between *P* and *E* in 1957 [11]. Following that, a justification to why there is a possibility for a stable solution with negative capacitances in the circuits appeared in 1976 [5].

**Figure 2 materials-12-03743-f002:**
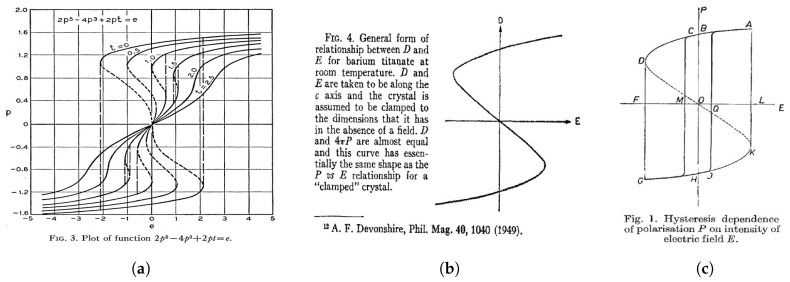
(**a**) Plots of the normalized polarization vs. the normalized electric field based on the theory developed by Devonshire and Slater from [12]. (**b**) Theoretical relation between the displacement field and the electric field in $BaTiO_{3}$ as shown in [11]. (**c**) Polarization vs. electric field relation as described in [13].

**Figure 3 materials-12-03743-f003:**
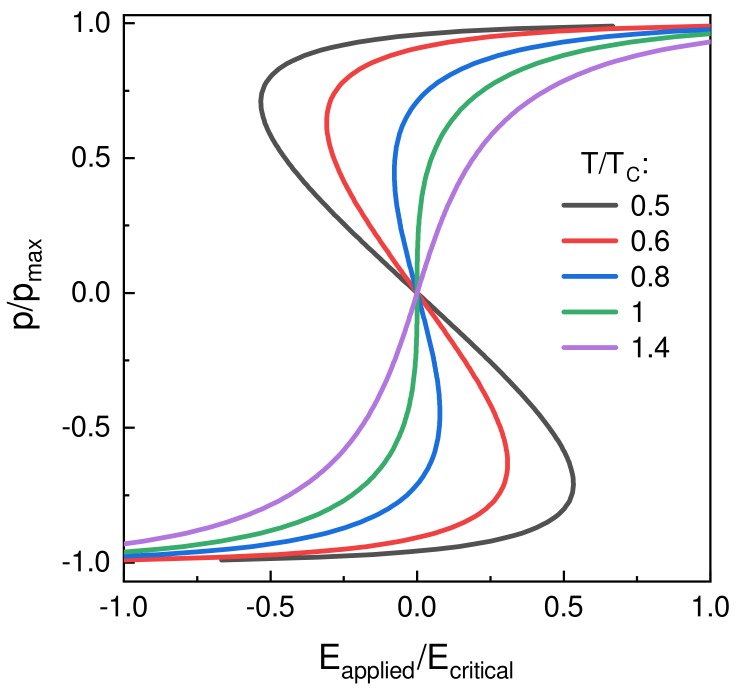
Theoretical relationship between the dipole moment *p* and the applied electric field $E_{applied}$ for different temperatures *T* normalized to the Curie-temperature $T_{C}$ according to Equation (Equation 11).

**Figure 4 materials-12-03743-f004:**
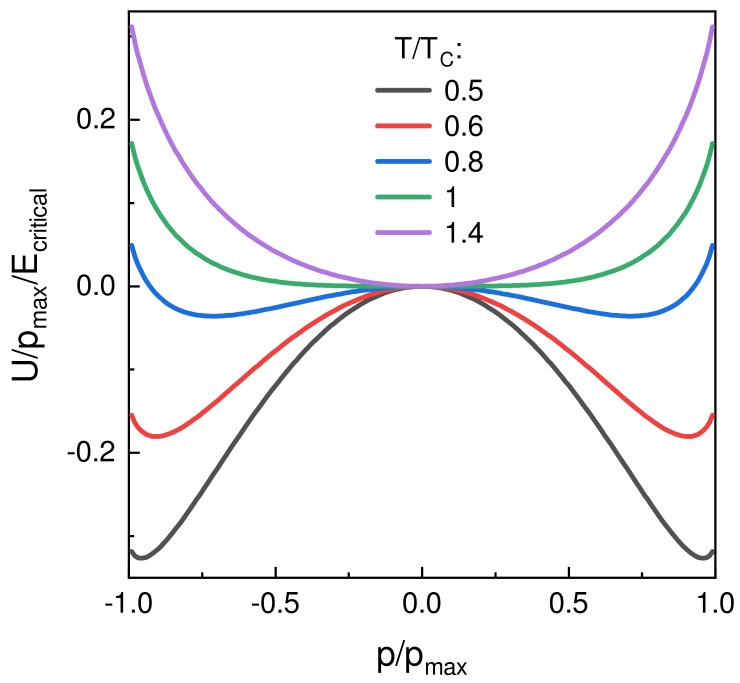
Theoretical relationship between the potential energy *U* normalized to $p_{max}$ and $E_{critical}$ vs. the dipole moment *p* for $T/T_{C}$ = 0.5, 0.6, 0.8, 1 and 1.4 using Equation (Equation 13).

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
