# Peer review of "Why Do Ferroelectrics Exhibit Negative Capacitance?"

_materials, 2019, doi:10.3390/ma12223743_

Round 1
Reviewer 1 Report
The authors discuss from a theoretical point of view the occurrence of negative capacitance, which seems to be an ubiquitous effect arising in ferroelectrics.
The manuscript is written very well and allows to gain insights into this interesting field of research. However, as this manuscript is denoted as a "review" the "get-home-message" is not straight forward to find. "This points to the fact, that the current understanding is not adequate ..." and "a great analysis ..." seem to be the final conclusion. The conclusion could be improved giving a more comprehensive overview of the direction of modelling, and describe at least briefly one possible experimental way of detecting negative capacitance.
Minor remarks:
1) page 3 "to simplify equation (3) the, a structural ..."
2) page 7 "it to decompose into decompose in ..."
3) there is a recent Nature Materials Review article "ferroelectric negative capacitance"
Author Response
Thank you for your comments and your time to read the manuscript in detail. The comments were very helpful. We have edited the manuscript to address your comments. Besides adding a reference to the mentioned Review article, we have also added another paragraph at the end of the conclusions as an outlook and to have a clearer take-home message. In terms of experimental methods for detecting negative capacitance, we feel that this is beyond the scope of this review, which focuses on the theoretical origin of negative capacitance. However, we have also added additional references to experimental works which will enable the reader to find additional information on these important topics.
Reviewer 2 Report
This review paper is expounded the negative capacitance from ferroelectrics based on a mean-field approach, as well as S-shaped polarization dP/dE < 0. One suggestion is practical example by experiment may more convincing for the readers, especially ferroelectric HfO2-based material.
Author Response
Thank you for your suggestion. We have added an additional paragraph at the end of the conclusions which discusses the use of HfO2-based ferroelectrics for negative capacitance devices.
Reviewer 3 Report
The negative capacity of the ferroelectric materials has been intensively investigated, both theoretically and experimentally. Over the past few years, many papers have discussed this topic:
1. J. Íñiguez, P. Zubko, I. Luk’yanchuk, A. Cano, Ferroelectric negative capacitance, Nature Reviews Materials volume 4, pages243–256 (2019)
2. Stuart Thomas, Negative capacitance found, Nature Electronics volume 2, page51 (2019)
3. Hyeon Woo Park Jangho Roh Yong Bin Lee Cheol Seong Hwang, Modeling of Negative Capacitance in Ferroelectric Thin Films, Adv.Mater., 31, 1805266 (2019)
4. Muhammad A. Alam, Mengwei Si, Peide D. Ye, A critical review of recent progress on negative capacitance field-effect transistors, Appl. Phys. Lett. 114, 090401 (2019)
In usual mode, scientists have used Landau’s theory of phase transitions to predict that an unstable negative capacitance region can arise in a ferroelectric material.
In the present paper "Why do ferroelectrics exhibit negative capacitance?" interesting is the approach of this subject. The model presented herein is overly simplified and ignores many of the complex interactions in real ferroelectrics. This study provides a simple, physical explanation of the negative capacitance phenomenon.
The paper presents an intuitive, microscopic description of the phenomenon of negative capacitance in ferroelectrics.
The manuscript can fit in "Materials" journal.
Introduction contents are correct and clear presented.
The "Model Description” section is clear and detailed presented.
Conclusions: the results of the study are synthesized in concordance with work.
This paper also cites suitable references, in comprehensive orders.
So I believe that the study will generate interests for the researchers in the field, and therefore this paper seems suitable for publishing in this journal.
Author Response
Thank you for your comments!
Reviewer 4 Report
The paper presents a comprehensive view of the origine of negative capacitance in ferroelectrics.
The paper must answer also to the following question ; Why is useful a negative capacitance and to present an overview of the negative capaictance transistors- a hot topic now.
Author Response
Thank you for your comments. We believe that several review papers and book chapters already exist that describe the usefulness of the concept in detail. We have now added some more references to such works (see below). We hope that the reviewer understands that the scope of this review was intentionally focused on the theoretical origin of negative capacitance, since this important topic has not been given much attention so far.
Khan, A.I., Energy-Efficient Computing with Negative Capacitance. In Advanced Nanoelectronics; John Wiley Sons, Ltd, 2018; chapter 7, pp. 179–200. Liao, Y.; Kwon, D.; Lin, Y.; Tan, A.J.; Hu, C.; Salahuddin, S. Anomalously Beneficial Gate-Length Scaling Trend of Negative Capacitance Transistors. IEEE Electron Device Letters 2019.
Alam, M.A.; Si, M.; Ye, P.D. A critical review of recent progress on negative capacitance field-effect transistors. Applied Physics Letters 2019, 114, 090401